# Exercise Dependence in Practitioners of Martial Arts and Combat Sports

**DOI:** 10.3390/ijerph192416782

**Published:** 2022-12-14

**Authors:** Karolina Kostorz, Wojciech J. Cynarski, Jacek Polechoński

**Affiliations:** 1Institute of Sport Sciences, The Jerzy Kukuczka Academy of Physical Education in Katowice, 40-065 Katowice, Poland; 2Institute of Physical Culture Studies, College of Medical Sciences, University of Rzeszow, 35-959 Rzeszow, Poland

**Keywords:** behavioral addiction, exercise addiction, hand-to-hand fights athletes, competition, physical activity

## Abstract

Background: The aim of this study was to analyse prevalence exercise dependence among practicing martial arts and combat sports. Methods: There were 166 respondents. The Exercise Dependence Scale—EDS was used. Results: The martial arts practitioners obtained a lower result in the ‘intention effects’ (*p* < 0.05; η^2^ = 0.03), ‘continuance’ (*p* = 0.04; η^2^ = 0.03), ‘lack of control’ (*p* < 0.05; η^2^ = 0.03), ‘reduction in other activities’ (*p* = 0.04; η^2^ = 0.03), and ‘total score’ (*p* = 0.04; η^2^ = 0.03) than the combat sports athletes. Both the respondents with a high training rank (*p* < 0.05) and subjects with above 5 years of training experience (*p* = 0.03; η^2^ = 0.03) achieved the higher mean in the ‘time’ subscale. Women obtained lower results in the case of ‘tolerance’ (*p* = 0.04; η^2^ = 0.04). The regression coefficient indicates that the higher respondent’s age, the lower total score she/he will achieve in the EDS. Conclusions: The findings have practical implications for identifying subjects ‘at-risk for exercise dependence’ symptoms, and may aid coaches and individuals in the implementation of a prevention program, to seek suitable support.

## 1. Introduction

The physiological, psychosocial, and cognitive benefits of regular physical activity (PA) are well documented [1,2,3,4,5,6]. On the other hand, some people can be involved in too much PA that results in negative health outcomes [7,8,9,10]. There are known cases of people in whom PA took a central place in life, eliminated other forms of spending free time, and regular training or exercise was undertaken even in the event of illness or injury [11,12,13,14,15]. These situations were considered pathological phenomena. Therefore, taking into account all of the negative effects, it is not surprising, that for several years, the issue mentioned above, has been the subject of interest of numerous authors [15,16,17,18,19,20]. Although there is no cut-off for “how much is too much”, most researchers agree that when exercise is excessive and is detrimental to both physical and mental health and becomes an obsession, it is defined by theorists as addiction behaviors [13,21,22,23,24,25]. However, it should be underlined that exercise dependence (or exercise addition) is not formally recognized within the Diagnostic and Statistical Manual for Mental Disorders (DSM-5-TR) [26].

Most often, exercise dependence is defined as a craving for leisure time PA that results in uncontrollable excessive exercise behaviour, the duration of the training with increasing frequency and severity, along with a person’s loss of self-control that demonstrates in physiological, psychosocial, as well as cognitive symptoms (e.g., fatigue, sudden heart attacks, injury, exercise-induced anaphylaxis, restlessness, eating disorders, insomnia, frustration, depression, tension, anxiety, conflicts with family members or friends, and social marginalisation) [16,17,18,27,28,29].

Several questionnaire tools have also been developed and validated to measure exercise dependence (e.g., the Obligatory Exercise Questionnaire developed by [30,31], as well as the Exercise Dependence Questionnaire worked out by Ogden, Veale, and Summers [32]). Most of the surveys show high validity and reliability [19,22]. Some authors used more than 1 questionnaire in their research to measure dependence [33,34]. According to the meta-analysis conducted by Nogueira et al. [35], the most often used are the Exercise Addiction Inventory—EAI [36] and the Exercise Dependence Scale—EDS [18]. However, the literature review revealed that the most frequently used measure was the Exercise Dependence Scale [37].

Recently, it is indicated that more attention should be paid to possible differences in exercise dependence prevalence among different sports [38], specifically in competitive disciplines [13]. The studies that fall under the General Theory of Martial Arts [39] have been carried out among men and women practicing combat sports [29,40], and among 25-year-old female martial artists [41]. However, taking into account the goals, tasks, and other aspects related to sports competition and recreational practice of traditional martial arts are divergent [42]. It was deemed reasonable to compare results obtained by martial arts practitioners and combat sport athletes. To the authors’ knowledge, no such studies have been undertaken so far. The willingness to fill the gap in scientific research was the inspiration to conduct our own investigations.

Therefore, the aim of the study was to examine prevalence of exercise dependence among people practicing combat sports and martial arts. Analyses were made with the consideration of the type of PA undertaken, the training rank and training experience, age, and sex of respondents. The following research questions were submitted:Are there statistically significant differences between the scores obtained by the combat sport athletes and martial arts practitioners?Do the factors of sex, training rank, and training experience determine the results of the studied variables?What percentage of respondents obtained results that classify them as asymptomatic-independent, symptomatic-independent, and at-risk of addiction to exercise?Is there a significant relationship between the three categories distinguished in the EDS, i.e., ‘at-risk for exercise dependent’, ‘non-dependent symptomatic’, and ‘non-dependent-asymptomatic’ groups, and the type of the PA undertaken, sex, training rank and training experience?Can the independent variables, i.e., the type of PA undertaken, age, sex, and training rank and training experience be used to predict a total score achieved in the EDS?

## 2. Materials and Methods

### 2.1. Procedure and Participants

The study was conducted electronically between December 2020 and April 2021. Respondents were recruited by advertising the survey on homepages of associations of different combat sports and martial arts, as well as social networks (e.g., Facebook). The subjects (n = 166) were of both sexes, at least 18 years old, and they had been training in martial arts (Pszczynska Martial Art, capoeira, aikido, kendo, wu shu) or combat sports (judo, wrestling, fencing, taekwondo, kyokushin karate) for at least 1 year. Based on the work of other authors [43,44,45] it was assumed that combat sport is any formula of competition derived from martial arts or ritual practices, related to direct or symbolic combat, in which the regulations have been institutionalized, and sports regulations created to protect the health and dignity of players. Whereas the martial arts are particular forms of physical or psycho-physical culture based on the traditions of warrior culture, which lead through the training of fighting techniques, to psycho-improvement and self-realization; these are simultaneous processes of education and positive asceticism. 

Data were collected from 166 participants. The study sample was comprised of 65 (39.16%) combat sport athletes and 101 (60.84%) martial arts practitioners. Women (n = 48) constituted 28.92% of the research group. Their mean age was 31.52 years (*SD* = 9.65). Men (n = 118) constituted 71.08% of the study group, with a mean age of 37.08 years (*SD* = 13.63). The respondents were divided into two groups depending on training experience: those who had been training for less (69 people) and for more than 5 years (97 subjects). The study also involved analyses depending on the participants’ training rank. It was assumed that a high rank meant having at least 3 kyu (3 kup for taekwondo athletes), and a white and orange colour for capoeira practitioners. In the case of wrestling and fencing, the rank of a player was determined by the coach, prior to the respondents completing the tool. In the statistical analyses, ‘high rank’ referred to respondents who obtained medal zone or placed 4–15 in their age categories. With these criteria, 69 subjects had a low rank and 97 respondents had a high rank. Table 1 contains detailed characteristics of the participants.

The study was conducted in accordance with the principles of the Declaration of Helsinki and according to the requirements of all applicable local and international standards. Participants were informed of the purpose of the study and electronically gave their voluntary informed consent to participate in the study and for the research results to be published in a scientific journal. People who agreed to participate in the study got the link to the questionnaire with a request to fill out the survey. Some questionnaires were not fully completed, and thus were excluded from further analysis. The participants had the option to withdraw from the study at any time without providing any reason for their decision.

Based on the data available on the website of the Central Statistical Office in 2019–2020, it was observed that about 129,106 people practise martial arts or combat sports in Poland, which constituted 11.7% of the total number of participants. A prior power analysis, conducted using G*Power 3.1.9.7 [46], ensured that sample sizes were sufficient to yield an adequate statistical power for the procedures conducted in the presented study. More specifically, to detect a significant finding (effect size f at the level of 0.25; alpha error probability at the level of 0.05) at a desired power level of 0.80, a total sample size of 128 participants was required. Therefore, it turned out that for the sample of 166 respondents, the margin of error was 7.6% and the confidence level above 80% was obtained.

### 2.2. Instruments

The diagnostic poll method with the questionnaire technique served to fulfil the assumed aims. A standardised research tool was applied. Exercise dependence was assessed with the Exercise Dependence Scale (EDS), developed by Hausenblas and Downs [18] in the version adapted by Danych, Polok and Guszkowska [47]. The tool contains 21 statements. Participants marked their responses to each of the 21-items in the blank space provided after each item. They marked their responses on a Likert scale anchored at the extremes with never (1) and always (6). A higher score reveals more exercise dependence symptoms. The EDS, which has seven dimensions for exercise dependence, was adopted from the DSM-IV criteria for substance dependence [48]. These are:(1)Tolerance is defined as either a need for increasing amounts of exercise to achieve the desired effects, or the diminished effect with continued use of the same amount of exercise (I continually increase my exercise intensity to achieve the desired effects/benefits).(2)Withdrawal is manifested by either the characteristic withdrawal symptoms for exercise or the same amount of exercise is engaged in to relieve or avoid withdrawal symptoms (I exercise to avoid feeling irritable).(3)Intention effects represent when exercise is often taken in larger amounts or more frequent than was intended (I exercise longer than I intend).(4)Lack of control is defined as the inability to stop or reduce your level of commitment to exercise (I am unable to reduce how long I exercise).(5)Time represents a great deal of time spent in activities that enable preparing for exercise (I spend a lot of time exercising).(6)Reduction in other activities assesses social, occupational, or leisure activities (hobbies) that are given up or reduced because of exercise (I would rather exercise than spend time with family/friends).(7)Continuance represents exercise that is continued despite being aware of a persistent or recurrent physical or psychological problems caused by the exercise (e.g., continued running despite injury) (I exercise despite recurring physical problems).

The total score of EDS is between a minimum of 26 and maximum of 126 points. Respondents who achieved scores in the range of 0 to 42 were classified as ‘non-dependent-asymptomatic’, while scores in the range of 43 to 84 were classified as ‘non-dependent-symptomatic’ and a cut-off score of 85 or more identified individuals considered ‘at-risk for exercise dependence’ [18,28]. Hausenblas and Downs [18] showed appropriate to high reliabilities for all scales( i.e., with the Cronbach’s alpha ranging from 0.67 ‘reduction in other activities’ to 0.93 for ‘withdrawal effects’). Measures used to assess the fit of the model to the data took acceptable values(i.e., Root Mean Square Error of Approximation—RMSEA = 0.06; Comparative Fit Index—CFI = 0.96; Tucker-Lewis Index—TLI = 0.95; Average Absolute Standardized Residua—AASR = 0.03; *p* < 0.05 [18]). These results are in line with research carried out by other authors [22,23,38,49,50,51,52,53,54,55]. In the presented research, indicators of the questionnaire’s reliability were obtained using Cronbach’s alpha, ranging from 0.70 for ‘reduction in other activities’ to 0.87 for ‘intention effects’ (Table 1). Therefore, it turned out that all scales had satisfactory reliability [56]. The survey required approximately 5 min to complete.

Participants also provided the following information: age, sex, rank, and training experience.

### 2.3. Statistical Analysis

Cronbach’s alpha coefficient was calculated to test the reliability of all subscales in the EDS. The basic analysis of the results was carried out using descriptive statistics (i.e., the mean (*M*), standard deviation (*SD*), median (*Me*), mode (*Mo*), coefficient of variation (*V*), and skewness (*As*)) and kurtosis (*Ku*) was calculated for the entire cohort, for the groups of martial arts practitioners and combat sports athletes, as well as for the population stratified by independent variables, i.e., sex, training rank and training experience. The analysis of normality of distribution was using the Shapiro–Wilk test. In turn, Levene’s test was used to access the homogeneity of variance. One-way analysis of variance was used to verify the significance of differences between the variables studied. As a post hoc test, the Tukey test for unequal groups was used. The significance level was assumed at *p* < 0.05. The effect size was also calculated Eta squared (η^2^) in each case, where statistically significant differences were found between the examined dependent variables. It was assumed that the effect size was small when the value of η^2^ was between 0.01 and 0.05, the medium between 0.06 and 0.13, and large above 0.14 [57]. The frequency tables allowed the assessment of the percentage of respondents who were asymptomatic-non-dependent, were symptomatic-non-dependent, and were at-risk for exercise dependence. Lastly, the stepwise multiple regression analysis was established to investigate whether the independent variables could be used to predict the dependent variable. In this case, the independent variables were age, sex, training rank and training experience, the type of PA undertaken, while the dependent variable was total score achieved in the EDS.

The analyses were performed with Microsoft Office Excel 2010 and StatSoft Statistica v. 13. Eta-squared was calculated using the IBM SPSS Statistics 27 program.

## 3. Results

It was found that the distribution of the studied variables was moderately asymmetric. All variables fell within the range of <−1 to 1>, for the whole sample, irrespective of the independent variables, i.e., the type of PA undertaken by the subjects, their sex, and training rank and training experience. It was also observed that the *Ku* values were satisfactory. *Ku* for all variables in each group fell within the <−1 to 1> range. All the results mentioned above are presented in Table 2.

In the first step of the analysis, the results of the martial arts practitioners were compared with the values obtained by the respondents training in combat sports (Table 3).

It turned out that the martial arts practitioners obtained statistically significant lower results in the case of following variables: ‘intention effects’ (*p* < 0.05; η^2^ = 0.03), ‘continuance’ (*p* = 0.04; η^2^ = 0.03), ‘lack of control’ (*p* < 0.05; η^2^ = 0.03), ‘reduction in other activities’ (*p* = 0.04; η^2^ = 0.03), and ‘total score’ (*p* = 0.04; η^2^ = 0.03) than the combat sport athletes. It should be added that in each case the effect size was small and equal to 0.03.

The following analyses considered the training rank of respondents (Table 4).

The collected data showed that respondents with a high training rank obtained a statistically significant higher result only for ‘time’ variable (*p* < 0.05). It was also observed that the effect size was small (η^2^ = 0.03).

To answer the question whether the results obtained in the EDS depended on the respondents’ training experience, analyses were carried out, the results of which are presented in Table 5.

The presented data show that the training experience differentiated only the one tested variable, i.e., ‘time’ (*p* = 0.03). It turned out that respondents with above 5 years of training experience obtained a statistically higher mean than individuals with a shorter sporting career and beginners. At the same it was observed that the effect size was small and equal to 0.03.

Subsequently, analysis was performed to determine whether the respondents’ results, obtained in the EDS, could be differentiated by sex (Table 6).

The analyses revealed that men achieved a higher mean in the case of ‘tolerance’ (*p* = 0.04). However, it was showed that the effect size was small and equal to 0.04.

The important issue was to assess the percentage of respondents who were ‘asymptomatic-non-dependent’, were ‘symptomatic-non-dependent’, and were ‘at-risk for exercise dependence’. For the purpose mentioned above, the data collected was tallied, organized, and its frequencies and percentages calculated and presented in the form of tables. A frequency table lists a set of values and how often each one appears. These tables help you understand which data values are common and which are rare. The results are presented taking into consideration the whole sample and in respect to the type of PA undertaken by the respondent, their sex, training experience, as well as training rank (Table 7).

According to the table presented above, it was found that 20 respondents (12.05%) were asymptomatic-non-dependent, 115 subjects (69.28%) were symptomatic-non-dependent, and 31 people from the entire sample (18.67%) were at-risk for exercise dependence. At the same time, it was revealed that there was no significant relationship between the three categories distinguished in the EDS (at-risk for exercise dependent, non-dependent symptomatic, non-dependent-asymptomatic groups), and the categorical variables such as the type of PA undertaken, their sex, as well as their training rank and training experience. In each case the *p* value was greater than alpha = 0.05.

The stepwise multiple regression analysis was established to investigate whether the independent variables, i.e., age, sex, and training rank and training experience, as well as the type of PA undertaken, can be used to predict the dependent variable which is the total score achieved in the EDS. Combat sports are defined as number ‘1′ and martial arts as number ‘0’. Table 8 summarizes the stepwise multiple regression.

It was shown that the type of PA undertaken, and age explained only 6% of the variance of the dependent variable, i.e., total score achieved in the EDS. The obtained values of the regression coefficient indicate that the higher the respondent’s age, the lower the total score she/he will achieve in the EDS. It also turned out that combat sport athletes probably achieved higher total scores in the EDS than martial arts exercisers. It was also observed that the Durbin–Watson test (DW) results (d) confirmed that there is a positive autocorrelation between the residuals of the model (DW = 1.90). Using the following DW critical value table (du and dl), by inputting sample size n, the number of regressors and the acceptable alpha level, it shows that d > du; therefore, there is no reason to reject the null hypothesis H(0), which means that first-order autocorrelation does not exist.

## 4. Discussion

It is indicated that despite the increased interest in exercise dependence, there is limited research examining the prevalence of this phenomenon in different sports disciplines, as well as limitations in the comparison of scores obtained by competitive athletes with scores achieved by non- competitive participants, i.e., people who exercise for recreational or for health purposes [13,25,29,35]. Taking into account all the aspects mentioned above, the aim of the study was to analyse the issue of exercise dependence among people participating in combat sports and martial arts. Analyses were made with the consideration of the type of PA undertaken, the training rank and training experience, the age, and respondents’ sex.

Firstly, it was found that martial arts practitioners obtained a statistically significant lower results in the case of ‘intention effects’, ‘continuance’, ‘lack of control’, ‘reduction in other activities’, and ‘total score’, compared to the combat sports athletes. These finding supports the view that competitive athletes tend to display more exercise dependence symptoms than non-competitive athletes, people practicing for recreational purposes [13,14,16,33,58,59,60,61]. According to Basoglu [62] the increase in the number of days and the number of hours of exercise seems to be a hallmark of exercise dependence. Based on these findings, it was possible to say that the ‘at-risk of exercise dependence’ was high in individuals who practice PA regularly, as part of a competitive sport [29]. In addition, the research conducted among triathletes [63] and marathon runners [59,64] showed a positive correlation between the number of weekly training hours and the ‘at-risk of addiction to exercise’.

Secondly, the collected data showed that respondents with a high training rank, as well as with above 5 years of training experience obtained statistically significant higher means for only one subscale, i.e., ‘time’. This finding suggests that the two independent variables mentioned above are not important moderator variables for exercise dependence symptoms. In turn, the results of the research conducted by Szabo et al. [65] indicate that the level of athletic training, and social context of the training affect exercise addiction, and, in line with the literature, the volume of exercise is not an index of susceptibility to exercise dependence. It is worth mentioning that using the EAI, Zeulner et al. [66] found no difference between results obtained by elite or amateur athletes. Youngman and Simpson [63] found no statistically significant association between the ‘at-risk for exercise addiction’ and the number of years of participating. On the one hand, it is suggested that training experience and number of weekly exercises are associated with the exercise dependency [62,67,68].

Thirdly, it was revealed that men obtained statistically significant higher means only in the case of ‘tolerance’. This finding is partly consistent with the results of other studies, which showed that the sex of competitive runners may not be a moderating factor for primary exercise dependence symptoms [61]. Subsequent research conducted by other authors [25,29,38,66,69] did not find any significant differences between men and women regarding exercise dependence symptoms. In turn, it was indicated that males tend to score higher than females on exercise dependence symptoms [14,16,17,21,22,38,70]. Szabo et al. [71] proved that EAI scores were higher in men than women. An important finding observed by Weik and Hale [34], is that men were significantly higher than women on the ‘withdrawal’, ‘continuance’, ‘tolerance’, ‘lack of control’, ‘time’, and ‘intention effect’ subscales EDS. At the same time, the results on the Exercise Dependence Questionnaire EDQ indicated that women scored significantly higher than men on the ‘interference’, ‘positive rewards’, ‘withdrawal’, and ‘social reasons’ subscales [34]. Statistical analysis using the *t*-test revealed that men had significantly higher total EDS scores than women, but women had significantly higher EDQ and Drive for Thinness (DFT) scores. The results are partly in line with the previous findings revealed by Zmijewski and Howard [72], who found that women are more dependent than men; with women scoring higher than men on weight control related subscales. In addition, Costa et al. [13] observed that the female athlete reported more ‘withdrawal effects’ symptoms than sportsmen. Interestingly, Cook, Hausenblas and Rossi [73] showed that men who were dissatisfied with their current weight reported more exercise dependence symptoms than women. Summarising all the results mentioned above, it can be stated that the evidence for sex differences is equivocal [34,35].

Fourthly, it turned out that 12.05% of respondents were asymptomatic-non-dependent, 69.28% of subjects were symptomatic-non-dependent, and 18.67% of individuals were ‘at-risk for exercise dependence’. The previous research has found varying prevalence rates for exercise dependence. Firstly, it should be mentioned that the study carried out by Orhan et al. [29], found that 3.5% of respondents were ‘asymptomatic-non-dependent’, 83.0% of subjects were ‘symptomatic-nondependent’, and 13.5% of individuals were ‘at-risk for exercise dependence’. In turn, Vardar et al. [40] did not observe the prevalence of exercise addiction among 11 athletes practicing judo, taekwondo, and karate. Furthermore, it showed that 7.8% of participants of the National Commando Training Center were ‘at-risk for exercise dependence’ [74]. Interestingly, those ‘addicted’ had a 1.53 times greater risk of injury during the commando course; however, none of those ‘addicted’ interrupted the course, compared to 25% of non-addicted participants injured.

It is also worth mentioning the results obtained by athletes from other disciplines. For example, Slay, Hayaki, Napolitano, and Brownell [75] showed that 26% of male runners and 25% of female runners were addicted to running, although subsequent research reported much higher rates. The analyses among triathletes revealed that between 20% [63] and 52% [76] met the criteria for exercise dependence. Interestingly, it turned out that training for longer distance races (i.e., the Olympics, Half-Ironman, and Ironman) put triathletes at a greater risk for exercise addiction than training for shorter races (i.e., Sprint) [63]. In turn, Allegre, Therme, and Griffiths [77] observed the lowest results and reported that 3.2% of ultra-marathon runners were addicted to exercise. These findings coincide with the results revealed by Freimuth et al. [78]. On the other hand, the subsequent research conducted by Szabo et al. [71] showed that the prevalence of ‘at-risk for exercise dependence’ was observed in 7–10% of university athletes and 17% among the ultra-marathon runners. In turn, the investigation conducted by Martin et al. [79] revealed that 44% of endurance runners were ‘at-risk for exercise addiction’. Besides, McNamara and McCabe [60] found that across 25 sports, 34.8% of Australian athletes were ‘at-risk for exercise dependence’. The investigation of exercise dependence symptoms among dancers, by contrast, revealed that 20.4% of participants were asymptomatic, 69.4% were symptomatic, and 10.2% had an exercise addiction. Moreover, those who were symptomatic (48.1%) and addicts (8.1%) were mostly Folk dancers [67]. In addition, Costa et al. [13] found, using the EDS, that among athletes, 18.3% were classified as ‘at-risk for exercise addiction’, 75.7% were classified as ‘non-dependent-symptomatic’, and 6.1% were classified as ‘non-dependent-asymptomatic’. In addition, analysis of sex and exercise dependence categories revealed significantly more athletes were in the nondependent-symptomatic category [13]. Moreover, a meta-analysis reveal that the EAI identified a higher proportion of people ‘at-risk for exercise addiction’ among endurance athletes (14.2%) followed by ball games (10.4%), fitness centre attendees (8.2%) and power disciplines (6.4%) [58].

At the same time, our own research revealed no significant relationships between the categories distinguished in the EDS and the categorical variables, i.e., the type of PA undertaken, their sex, as well as training rank and training experience. In turn, Orhan et al. [29] observed a significant difference between the asymptomatic group and other groups, which considered years of regular training. In addition, our findings are partly in line, especially regarding sex, with the results revealed by other authors [13,38,70,80]. In turn, Smith et al. [61] found a higher proportion of the competitive runners were classified as being ‘at-risk for exercise dependence’ compared to non-competitive runners. It should also be mentioned that Karademir [25] concluded that respondents with the least exercise addiction level were engaged in team sports.

Finally, the stepwise multiple regression analysis revealed that the type of PA undertaken, and age explained only 6% of the variance of the dependent variable, i.e., total score achieved in the EDS. The obtained values of the regression coefficient indicate that the higher the age of the respondent, the lower the total score she/he will achieve in the EDS. It also turned out that combat sport athletes probably achieved higher total scores in the EDS than a martial arts practitioner. Our findings, especially regarding age, are consistent with the results of other studies [13,81,82]. It is indicated that although exercise dependence afflicts subjects of all ages, it is considered more common among those younger than 35 years [22]. Costa et al. [22] indicate that the results they obtained both support previous research on the prevalence of exercise dependence and reveal that adulthood may be the critical age for developing exercise dependence. Although the authors mentioned above proved that the young adult group and the adult group reported higher levels of ‘tolerance’ and ‘time’ compared to middle-aged adults, middle-aged adults reported lower scores in the case of ‘reduction in other activities’ than the young adult group and lower scores in relation to ‘intention’ than the adult group, whereas no difference was observed in females with consideration of age groups [22]. It should also be mentioned that Bavlı et al. [67] in the study conducted among dancers showed that the symptomatic group had a statistically higher exercise age than those in the asymptomatic and addict groups, which had statistically higher daily exercise duration than the others. According to Szabo [83] the findings that the prevalence for exercise dependence decline with age may be due to the fact that the older exercisers develop a more balanced lifestyle. On the other hand, in the study on middle class weightlifters in relation to exercise dependence did not find differences between young adults (18–24 years) and adults (25–55 years) [84]. Regarding age, Orhan et al. [29] also failed to find a difference between results in the EDS obtained by combat sports athletes. In addition, the meta-analysis conducted by Nogueira et al. [35] revealed that many studies have failed to observe differences by age [63,64,69,85].

Therefore, in light of all the data mentioned above, it should be noted that the further studies on age differences for exercise dependence are required. It should also be underlined that the analyses of exercise addiction with the consideration of different participant ages is limited in the literature; therefore, it was not possible to accurately compare our findings with the results obtained by other authors.

## 5. Conclusions

Based on this study, it is possible to formulate the following conclusions:
The martial arts practitioners obtained lower results in the case of ‘intention effects’, ‘continuance’, ‘lack of control’, ‘reduction in other activities’, and ‘total score’ than the combat sport athletes, which was statistically significant.As compared with men, women achieved a lower mean in the case of ‘tolerance’, which was also statistically significant.The subjects with a high training rank obtained a higher scores only on ‘time’ subscale, showing statistical significance.The respondents with above 5 years of training experience achieved a statistically significant higher mean only related to ‘time’.It was found that 12.05% of respondents were asymptomatic-non-dependent, 69.28% of subjects were symptomatic-non-dependent, and 18.67% individuals were at risk for exercise dependence.The analyses revealed that there is no significant relationship between the three categories distinguished in the EDS (‘at-risk for exercise dependent’, ‘non-dependent symptomatic’, and ‘non-dependent-asymptomatic’ groups), and the categorical variables such as: the type of PA undertaken, their sex, as well as training rank and training experience.It was shown that the type of PA undertaken, and age explained 6% of the variance of the dependent variable, i.e., total score achieved in the EDS. The obtained values of the regression coefficient indicate that the higher the age of the respondent, the lower the total score she/he will achieve in the EDS. It also turned out that combat sport athletes probably achieved higher total scores in the EDS than martial arts practitioners.

The presented findings have practical implications for identifying individuals ‘at-risk for exercise dependence’ symptoms and may aid both coaches or instructors and the individuals in the implementation of prevention programs, seeking suitable support, as well as the treatment of exercise addiction. However, it should be noted that the Exercise Dependence Scale, which was used in this research, is a screening tool and not a diagnostic tool. Respondents classified as at-risk must undergo clinical interviews and/or medical exams to reliably assess exercise dependence [15].

### Limitations

Although the presented results advance the extant literature, design limitations exist. In addition, the potential areas for expanding the research should also be identified. First of all, it is advisable to involve many more respondents in further studies. Secondly, it should be underlined that the data provided was based on self-reporting and was therefore dependent on the respondent’s honesty and understanding of the questions asked. Although appropriate for evaluating subjective factors, future investigations could be supplemented with other tools (e.g., blood analysis of hormones—biomarkers of Relative Energy Deficiency in Sports—RED-S). Another very important aspect is that the respondents may obtain different results on the EDS, across a competitive season or significant moments in their life. Therefore, it seems that the scores for individuals with exercise dependence are not consistent but are characterised by certain dynamics and changeability in their lifetime. Therefore, it seems reasonable to perform longitudinal studies with the application of a cross-sectional and sequential analysis design, which also includes various psychological factors and, for example, seek a relationship between exercise dependence symptoms and weight gain, weight loss, or maintenance goals. It should also be underlined that the electronic survey was circulated during a period when most competitions and sports facilities were unexpectedly cancelled due to the COVID-19 pandemic. Another limitation of the study is that due to the use of various tools by researchers, it is impossible to fully compare the obtained results. Maselli et al. [38] emphasized that it is very hard to make consistent comparisons with studies that not only used different measures of exercise dependence, but also conceptualizations, and that some of the researchers did not operate a categorical subdivision of respondents into ‘dependent’ and ‘non-dependent’. Additionally, attention should be paid to the suggestions and conclusions established by Szabo [15], as the author emphasizes that the most frequently adopted path of research into exercise addiction may obscure ambiguous assumptions and one-sided quantitative analyses.

Taking into account all the limitations and aspects mentioned above, the obtained findings cannot be perceived as a final conclusion. The research presented here should constitute a starting point for further considerations.

## Figures and Tables

**Table 1 ijerph-19-16782-t001:** Demographic characteristics of the participants.

Research Group	Count	%	*M* Age	*SD* Age
Martial arts practitioners	101	60.84	36.77	12.06
Combat sports athletes	65	39.16	33.46	13.79
Women	48	28.92	31.52	9.65
Men	118	71.08	37.08	13.63
Respondents with high training rank	97	58.43	36.92	13.79
Respondents with low training rank	69	41.57	33.45	11.13
Respondents with > 5 years training experience	97	58.43	37.49	13.61
Respondents with < 5 years training experience	69	41.57	32.63	11.13

**Table 2 ijerph-19-16782-t002:** Descriptive statistics and Cronbach’s alpha values for the entire sample.

Studied Variable	*M*	*SD*	*As*	*Me*	*Mo*	*V*	*Ku*	*Cronbach’s Alpha*
Time	3.40	1.26	0.30	3.33	3.67	36.93	−0.62	0.84
Withdrawal effects	3.59	1.30	−0.14	3.67	4.00	36.27	−0.78	0.70
Intention effects	2.72	1.31	0.51	2.67	1.00	48.21	−0.37	0.87
Continuance	3.51	1.40	0.06	3.33	2.67	39.75	−0.89	0.73
Lack of control	3.05	1.28	0.16	3.00	2.67	41.77	−0.62	0.73
Tolerance	3.43	1.27	0.08	3.33	4.00	36.91	−0.65	0.82
Reduction in other activities	2.74	1.08	0.39	2.67	3.00	39.50	−0.08	0.70
Total score	67.36	67.00	0.16	19.27	multiple	28.60	−0.39	0.91

**Table 3 ijerph-19-16782-t003:** Comparison of results by between respondents practicing martial arts and combat sports.

Studied Variable	Combat Sports		Martial Arts		*F*	*df*	*p* Value	η^2^
	*M*	*SD*	*M*	*SD*				
Time	3.48	1.24	3.35	1.27	0.40	164	0.56	
Withdrawal effects	3.58	1.30	3.59	1.31	<0.01	164	0.99	
Intention effects	2.99	1.44	2.55	1.20	4.69	164	<0.05	0.03
Continuance	3.82	1.40	3.32	1.37	5.28	164	0.04	0.03
Lack of control	3.31	1.23	2.89	1.28	4.48	164	<0.05	0.03
Tolerance	3.68	1.25	3.28	1.26	4.01	164	0.07	
Reduction in other activities	2.97	1.22	2.59	0.96	4.99	164	0.04	0.03
Total score	71.52	19.86	64.68	18.48	5.11	164	0.04	0.03

*M*—mean; *SD*—standard deviation; *F*—result of the ANOVA; *df*—degree of freedom; *p* value—statistical significance level; η^2^—eta square.

**Table 4 ijerph-19-16782-t004:** Comparison of results by training rank of respondents.

Studied Variable	Low Rank		High Rank		*F*	*df*	*p* Value	η^2^
	*M*	*SD*	*M*	*SD*				
Time	3.15	1.19	3.57	1.28	4.60	164	<0.05	0.03
Withdrawal effects	3.76	1.26	3.46	1.32	2.08	164	0.18	
Intention effects	2.71	1.44	2.73	1.22	<0.01	164	0.95	
Continuance	3.42	1.42	3.58	1.38	0.53	164	0.50	
Lack of control	3.07	1.28	3.04	1.28	0.02	164	0.89	
Tolerance	3.47	1.37	3.41	1.20	0.09	164	0.78	
Reduction in other activities	2.78	1.19	2.71	1.01	0.16	164	0.71	
Total score	67.12	20.47	67.54	18.47	0.02	164	0.90	

**Table 5 ijerph-19-16782-t005:** Comparison of results by training experience of respondents.

Studied Variable	>5 Years of Training Experience		<5 Years of Training Experience		*F*	*df*	*p* Value	η^2^
	*M*	*SD*	*M*	*SD*				
Time	3.59	1.33	3.13	1.09	5.58	164	0.03	0.03
Withdrawal effects	3.43	1.30	3.81	1.27	3.60	164	0.08	
Intention effects	2.74	1.26	2.70	1.40	0.05	164	0.84	
Continuance	3.53	1.40	3.49	1.40	0.03	164	0.88	
Lack of control	3.02	1.29	3.11	1.26	0.20	164	0.68	
Tolerance	3.38	1.24	3.51	1.31	0.40	164	0.56	
Reduction in other activities	2.67	1.03	2.85	1.15	1.16	164	0.32	
Total score	67.06	18.96	67.78	19.83	0.06	164	0.83	

**Table 6 ijerph-19-16782-t006:** Comparison of results by sex of respondents.

Studied Variable	Women		Men		*F*	*df*	*p* Value	η^2^
	*M*	*SD*	*M*	*SD*				
Time	3.40	1.46	3.40	1.17	<0.01	164	0.99	
Withdrawal effects	3.83	1.36	3.49	1.27	2.46	164	0.19	
Intention effects	2.52	1.23	2.81	1.34	1.61	164	0.29	
Continuance	3.50	1.39	3.52	1.40	0.01	164	0.94	
Lack of control	2.95	1.37	3.10	1.24	0.44	164	0.58	
Tolerance	3.06	1.37	3.58	1.20	5.97	164	0.04	0.04
Reduction in other activities	2.67	1.18	2.77	1.04	0.33	164	0.63	
Total score	65.81	22.63	67.99	17.79	0.43	164	0.51	

**Table 7 ijerph-19-16782-t007:** The prevalence of the three categories of EDS among respondents.

Sample Characteristics		Exercise Dependence Symptoms						Pearson Chi-Square	*p* Value
		Non-Dependent-Asymptomatic		Non-Dependent-Symptomatic		At-Risk for Exercise Dependence			
		Count	%	Count	%	Count	%		
Whole Sample		20	12.05	115	69.28	31	18.67		
Type of the undertaken PA	Combat sports	7	10.77	41	63.08	17	26.15	3.94	0.14
	Martial arts	13	12.87	74	73.27	14	13.86		
Training experience	>5 years	11	11.34	69	71.13	17	17.53	0.38	0.83
	<5 years	9	13.04	46	66.67	14	20.29		
Training rank	Low rank	11	15.94	43	62.32	15	21.74	2.91	0.23
	High rank	9	9.28	72	74.23	16	16.49		
Sex	Women	10	20.83	28	58.33	10	20.83	5.66	0.06

**Table 8 ijerph-19-16782-t008:** Regression summary for the total score achieved in the EDS.

Regression Summary	R^2^ = 0.06; F(2.163) = 4.83; *p* < 0.01; Std.Error of Estimate: 18.84					
	b*	Std.Err.	b	Std.Err.	t (163)	*p* Value
Age	−0.16	0.08	−0.24	0.12	−2.11	0.04
Type of the undertaken PA	0.15	0.08	6.03	3.02	2.00	0.05

b*—BETA; b—regression coefficient; t—t-test result; 163—number of degrees of freedom; *p* value—statistical significance level.

## Data Availability

The data presented in this study are available on request from the corresponding author. The data are not publicly available due to they are collected, processed and calculated by the author.

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
