# Peer review of "Exercise Dependence in Practitioners of Martial Arts and Combat Sports"

_ijerph, 2022, doi:10.3390/ijerph192416782_

Round 1
Reviewer 1 Report
The aim of the study was to examine the prevalence of exercise dependence among people practicing combat sports and martial arts.
1. It is recommended to include in the abstract a brief description of the statistical analysis performed for a better understanding of the significance values ​​provided.
2. It is recommended to rearrange the information in the abstract since perhaps some information included in the results section can be included in the conclusions section.
3. It is recommended not to repeat in keywords the words already included in the title, for a better localization of the scope.
4. It is necessary to include the use of Cronbach's alpha in the "Statistical analysis" section.
5. It is necessary to include information in the table footers about the acronyms of the variables (it would even be advisable to highlight the values ​​with statistical significance in the table).
6. A better description of the title of Table 6 is recommended.
7. It is recommended to order the conclusions according to the research questions formulated in the introduction section.
8. Line 482: Include reference [15].
9. It is recommended to review the entire document and correct grammatical errors (eg, line endings, expressions, and words ("martials" on line 68, 280...), etc.
10. A revision and edition of the English language is recommended.
11. It is recommended to decide between the use of "sex" or "gender"; it is proposed to use "sex" since "gender" is a much more ambiguous concept today.
Author Response
We thank the referee for the careful and insightful review of our manuscript . Following the Reviewer’s suggestions, relevant changes have been made and any revisions to the manuscript were marked up using the “Track Changes”.
We appreciate for your warm work earnestly and hope that the correction will meet with approval.
Reviewer:
- Although I appreciated the content of the manuscript, in my opinion, it is difficult to see how it significantly adds new information to the literature. The manuscript would benefit from a more precise narration, I would suggest reorganizing the discussion, making it more concise and clearer and the conclusions.
We kindly draw the reviewer’s attention that to our knowledge, there are no data based including similar problems. The own research is the first study in which it was analyzed prevalence exercise dependence among practicing martial arts and combat sports with the consideration of the type of the undertaken PA, the training rank and experience, as well as the age and the respondents’ sex. It was an important step in understanding the phenomenon and the differences in the symptoms of exercise dependence among types of the undertaken PA, which was the mentioned above.
We tried to reorganizing and shorting the discussion. We also ordered the conclusions according to the research questions formulated in the introduction section.
These changes are presented in the manuscript in red given the volume of additions.
Reviewer:
- There are some typos in the manuscript.
Thank you for your comments. The English has been carefully polished and the revisions are highlighted in red. The manuscript was carefully and insightfully check by the recognized sworn translator. We believe that the manuscript will receive approval from the Editors.
Reviewer:
- I suggest summarizing the characteristics of the participants in a table.
Thank you for your comments. We have added a table summarizing the characteristics of the participants.
Reviewer:
- I suggest creating a graphical abstract with the purpose, study design and results to help the reader quickly gain an overview of the article.
Thank you for your suggestion, but in our opinion, it is unnecessary, because the research questions are bulleted and highlighted in the text, which allows the reader to quickly get to know the subject matter of the article.
We appreciate for your warm work earnestly and hope that the correction will meet with approval.

Reviewer 2 Report
In this study authors analyzed the prevalence of exercise dependence among practicing martial arts and combat sports.
Here are some comments to improve the quality of this article:
1. Although I appreciated the content of the manuscript, in my opinion, it is difficult to see how it significantly adds new information to the literature. The manuscript would benefit from a more precise narration, I would suggest reorganizing the discussion, making it more concise and clearer and the conclusions.
2. There are some typos in the manuscript.
3. I suggest summarizing the characteristics of the participants in a table.
4. I suggest creating a graphical abstract with the purpose, study design and results to help the reader quickly gain an overview of the article.
Author Response
We thank the referee for the careful and insightful review of our manuscript . Following the Reviewer’s suggestions, relevant changes have been made and any revisions to the manuscript were marked up using the “Track Changes”.
We appreciate for your warm work earnestly and hope that the correction will meet with approval.
Reviewer:
- Although I appreciated the content of the manuscript, in my opinion, it is difficult to see how it significantly adds new information to the literature. The manuscript would benefit from a more precise narration, I would suggest reorganizing the discussion, making it more concise and clearer and the conclusions.
We kindly draw the reviewer’s attention that to our knowledge, there are no data based including similar problems. The own research is the first study in which it was analyzed prevalence exercise dependence among practicing martial arts and combat sports with the consideration of the type of the undertaken PA, the training rank and experience, as well as the age and the respondents’ sex. It was an important step in understanding the phenomenon and the differences in the symptoms of exercise dependence among types of the undertaken PA, which was the mentioned above.
We tried to reorganizing and shorting the discussion. We also ordered the conclusions according to the research questions formulated in the introduction section.
These changes are presented in the manuscript in red given the volume of additions.
Reviewer:
- There are some typos in the manuscript.
Thank you for your comments. The English has been carefully polished and the revisions are highlighted in red. The manuscript was carefully and insightfully check by the recognized sworn translator. We believe that the manuscript will receive approval from the Editors.
Reviewer:
- I suggest summarizing the characteristics of the participants in a table.
Thank you for your comments. We have added a table summarizing the characteristics of the participants.
Reviewer:
- I suggest creating a graphical abstract with the purpose, study design and results to help the reader quickly gain an overview of the article.
Thank you for your suggestion, but in our opinion, it is unnecessary, because the research questions are bulleted and highlighted in the text, which allows the reader to quickly get to know the subject matter of the article.
We tried our best to improve the manuscript and made some changes marked in red in revised paper.
Once again, thank you very much for your comments and suggestions and and we hope the correction will meet with approval.
